# Physical and Mechanical Properties of Ammonia-Treated Black Locust Wood

**DOI:** 10.3390/polym12020377

**Published:** 2020-02-07

**Authors:** Mariana Domnica Stanciu, Daniela Sova, Adriana Savin, Nicolae Ilias, Galina A. Gorbacheva

**Affiliations:** 1Department of Mechanical Engineering, Faculty of Mechanical Engineering, Transilvania University of Brașov, B-dul Eroilor 29, 500036 Brașov, Romania; sova.d@unitbv.ro; 2National Institute of Research and Development for Technical Physics, B-dul Mangeron 47, 700050 Iasi, Romania; asavin@phys-iasi.ro; 3Faculty of Mining, University of Petrosani, 20 Universității Street, 332006 Petroșani, Romania; iliasnic@yahoo.com; 4Technical Sciences Academy of Romania, B-dul Dacia 26, 030167 Bucuresti, Romania; 5Faculty of Forestry, Forest Harvesting, Wood Processing Technologies and Landscape Architecture, Mytishchi Branch of Bauman Moscow State Technical University, 1st Institutskaya street, 141005 Mytischi, Russia; gorbacheva-g@yandex.ru

**Keywords:** black locust wood, ammonia treated wood, colour change, dynamic mechanical analysis

## Abstract

Because of the uneven colour of black locust wood, different technologies are used to change the colour, the bestknown being chemical and thermal treatments. Some of them affect the mechanical properties of wood, such as elasticity modulus, strength, durability. This study aims to compare the physical and mechanical properties of black locust wood control samples and treated wood samples with ammonia hydroxide, in terms of density profile, colour values (CIE *L**, *a**, *b**), mechanical properties of samples subjected to static bending, viscous-elastic properties (storage modulus (*E*’), loss modulus (*E*”) and damping (tanδ)). Two types of ammonia-fuming treatment were applied on samples: first treatment T1-5% concentration of ammonia hydroxide for 30 days; second treatment T2-10% concentration for 60 days. The results highlighted the following aspects: the overall colour change in the case of the second treatment is 27% in comparison with 7% recorded for the control samples; the lightness and yellowness values are the most affected by the second ammonia treatment of black locust wood. The density increased with almost 20% due to ammonium fuming (10% concentration/60 days); in case of static bending, the elastic modulus (MOE) tends to decrease with increasing the exposure time to ammonium, but the modulus of rupture (MOR) increases with almost 17% and the breaking force increases too, with almost 41%. In the case of dynamic mechanical analysis, the temperature leads to different viscous-elastic behaviour of each type of samples.

## 1. Introduction

The physical and mechanical properties of wood are very important for its use. The black locust wood (Latin name *Robinia pseudoacacia*) has different applications, like fence posts, boatbuilding, flooring, furniture, mine timbers, railroad ties, turned objects, fingerboards and back plate of guitars, some parts of musical instruments, veneer, wooden garden benches, pergolas etc. [1]. Due to the anatomical and chemical structure, the black locust wood is characterized by high natural durability, a fact for which it is used mainly for outdoor products. For the product diversification, harmonizing the colour of the wood with the architecture of the space and increasing the resistance to environmental factors, the colour of the black locust wood is modified by various treatments, but the most significant are chemical treatments. Chemical and thermal modifications can lead to improved wood characteristics, such as its durability, but also to the elastic properties modification, either by reducing or improving the mechanical strength. The surface modification of wood by different technologies improves the UV stability and change the surface energy of wood [2]. According to the macro and microscopic structure, natural wood is a composite material consisting of fibers and the matrix(lignin) [3,4,5]. In this context, Weigl et al. reported the influence of ammonia treatment on different wood species related to the modification of their physical properties [6]. They also noticed that one of the effect of ammonia treatment is the increase of wood affinity to water, but also the color modification and stabilization are enhanced [6]. The most receptive to ammonia-fumed coloring is black locust wood. In [7,8], the authors reported that the colour of ammonia-fumed oak is more resistant than the heat treated oak after UV exposure. Also, in [8] the authors studied the influence of wood species, time of exposure to ammonia gas on colour change of wood, using Fourier Transform Infrared (FTIR) spectroscopic analysis. The most significant chemical modification is related to the reaction of ammonia with carboxylate groups from wood with formation of ammonium salt, the reaction with aldehyde and ketonic groups resulting imines and the reaction with ester functional groups leading to formation of amides [9]. In [10], the authors studied the density profile and thermal properties of untreated beech wood and treated with ammonia and compressed beech wood. Many researchers investigated the effect of ammonia treatment on oak wood, both from colour modification and mechanical properties points of view [6,7,9,10]. The uneven colour of the natural black locust wood is an aesthetic disadvantage of this species and therefore, the steaming of the black locust wood and fuming with ammonia gas for the uniformity of the colour, is a widespread practice in some countries. In [11,12], the authors reported the significant effect of black locust wood treated with ammonia on shear strength in case of different glues, in comparison with untreated wood. Kačík et al. [12] evaluated the modification of chemical components of black locust wood after hot-water pretreatment, highlighting that the most affected by the treatment are lignin, hemicelluloses and holocelluloses [12].

Even if there are numerous studies on the colour changes of wood exposed to ammonia gas, there is some lack of information about the correlation between viscous-elastic properties of black locust wood, density profile and colour modification parameters. The chemical reactions have led to mechanical properties modification. The aim of the study is to analyze the physical and mechanical properties of black locust wood, both untreated and treated specimens, with solution of ammonium hydroxide, knowing that the darker wood colour in ammonia fuming is accomplished through chemical reactions between ammonia gas and wood compounds, which can affect the properties of black locust wood.

## 2. Materials and Methods

### 2.1. Materials

Three kinds of samples were tested: control samples-without treatment, cut in radial–longitudinal direction (S_R_T_0_) and cut in semi-radial–longitudinal direction (S_SR_T_0_); samples submitted to treatment T1 (surface treatment), cut in radial–longitudinal direction (S_R_T_1_) and cut in semi-radial–longitudinal direction (S_SR_T_1_); samples submitted to treatment T2 (in-depth treatment), cut in radial–longitudinal direction (S_R_T_2_) and cut in semi-radial–longitudinal direction (S_SR_T_2_).24 samples of black locust wood were investigated, four from each type. The treatment of wood samples consisted in the exposure of wood to fumes of ammonium hydroxide (T1-5% concentration and 30 days; T2-10% concentration and 60 days). Ammonium hydroxide reacts with the wood tannins. After the treatment, the samples were dried in an oven chamber to obtain the moisture content range between 7% and 8%, specific for some parts of musical instruments construction. The physical and geometric characteristics of the wood samples are indicated in Table 1. The wood specimens with sizes 50 × 10 × 5 mm^3^ were subjected to bending, the 6 N force being applied in the middle of the distance between supports (40 mm). In Figure 1, the specimens tested for dynamic mechanical analysis (DMA) and colour change assessment are presented.

### 2.2. Methods

The experimental investigations consisted of density profile analysis (DPA) on samples subjected to the same treatment as the other ones subjected to colour measurement (CM), a static three-point bending testand dynamic mechanical analysis (DMA). Finally, from the samples subjected to DMA, there were prepared some specimens which were covered with thin gold particles for scanning electron microscopy.

#### 2.2.1. Density Profile Analysis (DPA)

The density profile of black locust samples on transversal direction and radial longitudinal direction was determined using an X-ray Density Profile Analyzer DPX300. The samples with the dimensions 50 × 50 × 30 mm^3^ were automatically weighed and tested by the equipment devices. Then, each specimen was introduced in the X-ray device train where the equipment measured the density profile using the X-ray flux. Figure 2 presents the steps followed during the DPA.

#### 2.2.2. Colour Measurement (CM)

In order to find out the colour variation of black locust wood affected by chemical treatment with ammonium hydroxide, the chroma meter CR-400 Konica Minolta was used. Measuring results were colour values using the *L***a***b** colour system, where *L** describes the lightness, and *a** and *b** describe the chromatic coordinates on the green–red and blue–yellow axes. The overall colour change Δ*E** was calculated with relations (1)–(3) [7]:(1)$$\Delta E_{T0-T1}^{*}=\sqrt{{\left(L_{0}^{*}-L_{1}^{*}\right)}^{2}+{\left(a_{0}^{*}-a_{1}^{*}\right)}^{2}+{\left(b_{0}^{*}-b_{1}^{*}\right)}^{2}},$$
(2)$$\;\Delta E_{T1-T2}^{*}=\sqrt{{\left(L_{1}^{*}-L_{2}^{*}\right)}^{2}+{\left(a_{1}^{*}-a_{2}^{*}\right)}^{2}+{\left(b_{1}^{*}-b_{2}^{*}\right)}^{2}},$$
(3)$$\Delta E_{T0-T2}^{*}=\sqrt{{\left(L_{0}^{*}-L_{2}^{*}\right)}^{2}+{\left(a_{0}^{*}-a_{2}^{*}\right)}^{2}+{\left(b_{0}^{*}-b_{2}^{*}\right)}^{2}}.$$
where $\Delta E_{T0-T1}^{*}$ represents the overall colour change between samples subjected to treatment T1 and untreated samples (T0); $\Delta E_{T1-T2}^{*}$–the overall colour change between samples subjected to treatment T2 and samples subjected to treatment T1; $\Delta E_{T0-T2}^{*}$–the overall colour change between samples subjected to treatment T2 and untreated samples (T0); $L_{0}^{*},\;a_{0}^{*},\;b_{0}^{*}$–colour system in terms of lightness, greenness and yellowness of control (untreated) samples; $L_{1}^{*},\;a_{1}^{*},\;b_{1}^{*}$–lightness, greenness and yellowness of treated samples subjected to the first treatment (5% ammonia);$L_{2}^{*},\;a_{2}^{*},\;b_{2}^{*}$–lightness, greenness and yellowness of treated samples subjected to the second treatment (10% ammonia).

#### 2.2.3. Bending Tests

(1). Static Three-Points Bending Test

The first step was the three-points bending test performed on three types of samples, ten samples for each type, by using the universal machine LS100 Lloyd’s Instrument with the load capacity of 100 kN. The aim of this test was to determine the breaking force, elasticity modulus of bending (MOE)and modulus of rupture (MOR) of black locust wood samples for untreated, T1-treated and T2-treated specimens with similar dimensions (length × width × thickness: 150 mm × 10 mm × 6.5 mm) like those of the samples subjected to DMA. The moisture content of wood (MC) was 6–8%, the environmental temperature T = 22 ± 1 °C and the relative humidity of air RH = 50% ± 5%. In Figure 3a, the principles of the three-point bending test can be observed and in Figure 3b, the breaking of black locust wood sample during the bending test. The speed of loading was set at 5 mm/min and the span between supports was 64 mm.

(2). Dynamic Mechanical Analysis (DMA)

The method of dynamic mechanical analysis (DMA) consists in applying an oscillating force at different frequencies (*f* = 1, 3.3, 5, 10, 50 Hz) in two cases. Firstly, isothermal conditions were used (temperature was kept constant at 30 °C during the test). Each sample was subjected to five DMA procedures with different load frequencies. The second analysis consists in the variation of temperature between 30 and 120 °C for 45 min, being repeated for different loading frequencies at the same values as in the first tests. The device returns the response of the material as a function of temperature and frequency that depends on the viscous-elastic nature of the material. The storage modulus (*E*’), the loss modulus (*E*’’) and the complex modulus (*E**) are calculated from the material response to the sine wave. The ratio of the loss modulus and the storage modulus is called damping, denoted by tanδ, which represents the capacity of the material to store strain energy. This type of analysis predicts the flow behavior of wood in different environmental conditions. In Figure 4, the principles of samples loading and the main components of the equipment are shown. Figure 4a presents the direction of wood grain related to loading: the force is applied perpendicularly to the longitudinal axis (denoted L or x) producing a radial or semiradial bending moment (denoted by M_R_ and M_SR_). In Figure 4b, the principles of the three-point bending test are shown and in Figure 4c, the main parts of the DMA equipment and the position of the wood sample within the device.

#### 2.2.4. Scanning Electron Microscope Hitachi S3400N

The microscopic views of black locust wood were captured with a Hitachi S3400N scanning electron microscope (SEM). Before SEM analysis, the small pieces of wood samples were coated with a thin layer of gold (Au), which is a conducting material, as can be seen in Figure 5. The SEM tests were performed in semi-vacuum conditions.

## 3. Results

### 3.1. X-ray Density Profile Analyzer DPX300

Density is correlated with the physical characteristics of the wood species and with their mechanical properties. Figure 6 presents the density profile of black locust samples in longitudinal–radial direction (Figure 6a) and in transversal direction (radial–tangential plane) (Figure 6b). The average values of density determined at 8%–10% moisture content, ranged between 700 and 780 kg/m^3^, values that are similar to those reported in literature [13,14,15]. It is known that black locust wood is characterized by a complex and uneven structure, with clearly visible annual rings. Since black locust wood is a deciduous ring porous species with wide early wood vessels, heavily clogged by tyloses [16], the density profile varies on the annual ring width depending on the areas with early and late wood, as can be seen in Figure 6b. There is a slight increase in density of ammonia-treated black locust wood in comparison with untreated black locust wood, but the values do not exceed the average values recorded in the literature for black locust wood. Density determined by using the X-ray method is almost 7% higher than the calculated one (from specific gravity relation as ratio between mass and volume of samples), as can be seen in Figure 7.

### 3.2. Colour Measurement

Figure 8a shows the influence of ammonia treatment on black locust wood from lightness *L**, chroma from green to red (*a**) (Figure 8b) and chroma from a blue to yellow (*b**) point of view (Figure 8c). The major colour changes were recorded in case of lightness: after the first treatment, the lightness decreased with 10% and after the second treatment, the differences were about 37% in comparison with the control samples. The overall colour change $\Delta E_{T0-T2}^{*}$ of the samples after the second treatmentwas around 27.073 units, as compared to the overall colour change $\Delta E_{T0-T1}^{*}$ obtained after the first treatment, whose value was 7.108 units. The overall colour change value $\Delta E_{T1-T2}^{*}$ obtained after the second treatment in relation to the first ammonia exposure was 20.379 units, which means that the period of exposure to ammonia (60 days) and the concentration of ammonia (10%) have the greatest influence on the colour change.

### 3.3. Bending Test

#### 3.3.1. Static Three-Point Bending Test

The static three-point bending test has revealed that the MOE decreases with almost 11% in case of treated samples, as compared to control samples, but the breaking force increases with 21% (for samples T1) and with 41% for the second treatment. Also, MOR increases for ammonia treated samples with 15%–17% in comparison with control samples. Figure 9 shows the characteristic curves for each type of tested samples. In Table 2, the average values and standard deviation of the main mechanical properties of control and treated samples are presented. The similar percentile changes were obtained by Weigl et al. [6], Čermák and Dejmal [10], who considered the reduction of mechanical properties values not significant. On the other hand, Rousek et al. [17] studied the possibilities of mechanical properties improvement in case of beech wood modification with ammonia gas, reporting a tendency to increase the mechanical properties of beech wood treated with ammonia, especially the MOR to compression.

#### 3.3.2. Dynamic Mechanical Analysis (DMA)

The experimental investigations have resulted in numerous data about the viscous-elastic behavior of different tested species. Thus, the first analysis consists in applying the load at different frequencies (1, 3.33, 5, 10, 50 Hz) at constant temperature (T = 30 °C),being determined the storage modulus values *E*’, loss modulus values *E*” and the damping tanδ. During the cyclic loading, wood tends to store increasingly more energy due to internal friction occurring in wood, as shown in Figure 10, but the damping capacity of wood decreases over time, regardless of species. Generally, the ability of wood samples to store energy increases with increasing the time of loading. This trend is similar with the strain hardening of steel, which is the process of making a metal harder and stronger through plastic deformation. The second analysis consists of temperature scanningat different frequencies. 

(1) Isothermal Conditions (T = 30 °C)

It can be seen that the elastic (storage) modulus *E*’ tends to increase by 6%–7%, thus increasing the time of loading. By increasing the frequencies (from 1 to 50 Hz), the storage modulus increases by almost 2.4% (Figure 10). The treatment of the dried black locust samples with ammonia leads to the increase of the storage modulus: after the first treatment, the increase is 4.8%–5% and after the second treatment, the increase is 50%, as compared to the control samples (Figure 10a–c). This phenomenon is explained by the effect of ammonia evaporation that leads to the increase of wood stiffness [16,17]. It is worth to mention that the black locust wood samples treated with ammonia were conditioned and dried to 6%–8% moisture content. Thus, the plasticization effect of ammonia vapor was eliminated. An improvement of mechanical properties of ammonia treated samples in case of beech wood was reported by [18] who noticed the differences between the treatment with ammonia gas (dry wood) and ammonia with water (wet wood).

The values of storage modulus ranged between 8800 MPa for untreated samples, 9400 MPa for the first treatment of black locust samples and 14,000 MPa for the second treatment (Figure 11). Similar values are reported by other authors [16,19,20,21]. The ratio between loss modulus and storage modulus represents damping (tanδ), which is a sensitive indicator of the mechanical or thermal conditions during the mechanical energy input dispersed as heat by internal friction caused by chain motion. The damping tends to decrease by increasing the loading time at a constant temperature of 30 °C. Because wood is a natural polymer and has a stratified structure of early and late wood, damping occurs gradually, in stages. The energy dissipates progressively, the cellular voids leading to the damping of the internal wood friction. This phenomenon is observed in form of variation curves of damping in Figure 12. In case of the samples T2 (Figure 12c) it can be noticed that the slope of the curve shows a decreasing linear variation over time, which can be influenced by chemical modification of wood by formation of different chemical groups.

(2) Temperature Scanning

Temperature affects the stiffness and resilience of wood characterized by thermoset behaviour. In the case of the control samples, the storage modulus increases slightly up to 75–80 °C, then the recording of a decreased trend (Figure 13, a—red lines) is noticed; in the case of treated samples (T1—blue lines and T2—black lines), the storage modulus has the trend to remain constant between 30–65 °C, after that, a trend to increase between 65 and 100 °C can be seen (Figure 13). 

The cross-section of wood and the relative position of loading with respect to grains (radial and semi-radial) influence the mechanical response of samples: for untreated samples and samples T1, the trend is similar with that of the samples cut in radial direction, but the samples T2have shown the tendency of decreasing the storage modulus (*E*’) starting with the temperature of 50 °C. The overshoot (peak) recorded for both treated samples, at the loading frequency of 50 Hz, is caused by molecular rearrangements that occur due to the increased free volume at the transition [22]. With increasing the temperature, the loss modulus *E*’ increases too for all types of samples; at temperatures higher than 100 °C, the values are double (Figure 14). An interesting behaviour regarding the variation of the loss modulus with temperature is recorded by semi-radial ammonia treated samples using the second treatment T2, which tends to decrease starting with 48 °C (Figure 14b). The viscous-elastic behavior is influenced by temperatures higher than 48 °C, also revealed in damping variation (Figure 15).

### 3.4. Scanning Electron Microscope Hitachi S3400N

In Figure 16 and Figure 17, the SEM capture of untreated and ammonia treated black locust wood samples is presented. At microscopic level, no differences between the three types of samples can be observed. The control samples cross-section shows thatthe wide early wood vessels (150–220 μm) are arranged in a 2–3 vessels-thick ring and are heavily clogged by tyloses. Latewood vessels have smaller diameters (70–140 μm). The longitudinal and the ray parenchyma often contain crystalline deposits. Molnar et al. and Nemeth et al. presented numerous studies on microstructure of black locust wood, highlighting the microscopic characteristics [20,21].

## 4. Conclusions

The paper presents the experimental results of numerous studies on black locust wood samples, uncoloured and coloured, as a result of exposure to ammonium hydroxide.

The overall colour change in the case ofthe second treatment is 27% in comparison with 7% recorded for the control samples. The lightness and yellowness are the most affected colour values after the second treatment of black locust wood with ammonia hydroxide (the lightness decreased with almost 40% after ammonia treatment and the yellowness—with 50%).The density increased with almost 20% due to ammonium fuming (10% concentration/60 days).At the static three-point bending, MOE recorded a decrease with almost 11% and MOR increased for ammonia treated samples with 15% to 17% in comparison with control samples.In the DMA test, the storage modulus (*E*’) is higher with almost 60% in the case of the second treatment of samples in comparison with control samples.The viscous behaviour is more evident when the temperature increases above 40–60 °C for all types of samples.The exposure of treated samples (T2) to temperature led to different behaviours according to the direction of wood grain related to the orthogonal directions of the samples (radial versus semiradial).The microscopic views captured with SEM did not highlight a specific surface modification of wood.

## Figures and Tables

**Figure 1 polymers-12-00377-f001:**
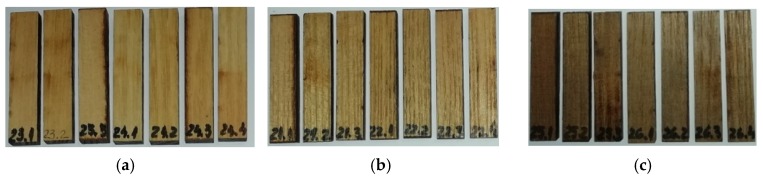
The wood samples: (**a**) Control sample (without treatment); (**b**)T1-samples subjected to 1^st^ treatment (5% ammonium hydroxide); (**c**) T2-samples subjected to 2^nd^ treatment (10% ammonium hydroxide).

**Figure 2 polymers-12-00377-f002:**
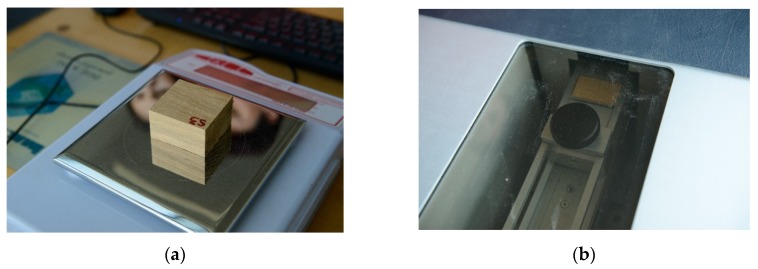
X-ray Density Profile Analyzer DPX300: (**a**)weighting the samples; (**b**)the samples in the X–ray device train.

**Figure 3 polymers-12-00377-f003:**
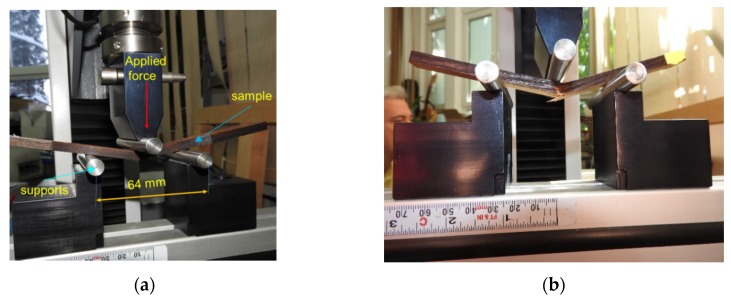
Three-point bending test: (**a**) the principle of loading; (**b**)the breaking of samples.

**Figure 4 polymers-12-00377-f004:**
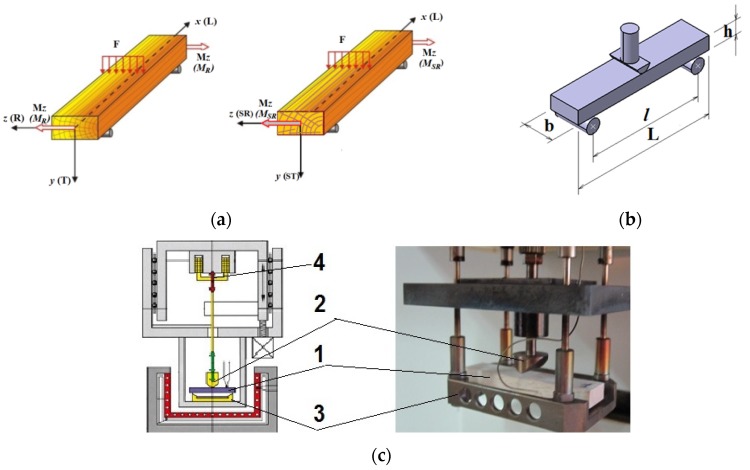
The DMA experimental set-up: (**a**) the types of wood samples (L –longitudinal direction, along wood grain; R–radial direction; T–tangential direction; SR—semi-radial direction; ST–semi-tangential direction); (**b**) the sample position on the equipment supports; (**c**) the main parts of DMA 242C Netzsch equipment: 1–specimen, 2–force application device, 3–specimen fixing/support devices, 4–electronic system for cyclic force application.

**Figure 5 polymers-12-00377-f005:**
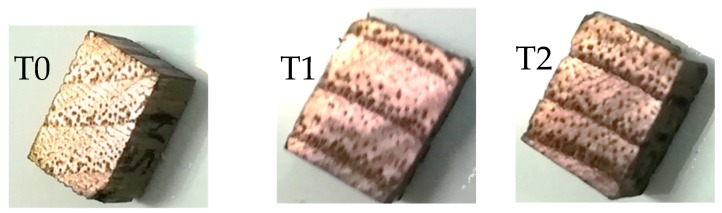
The gold-coated samples prepared for scanning electron microscope (SEM).

**Figure 6 polymers-12-00377-f006:**
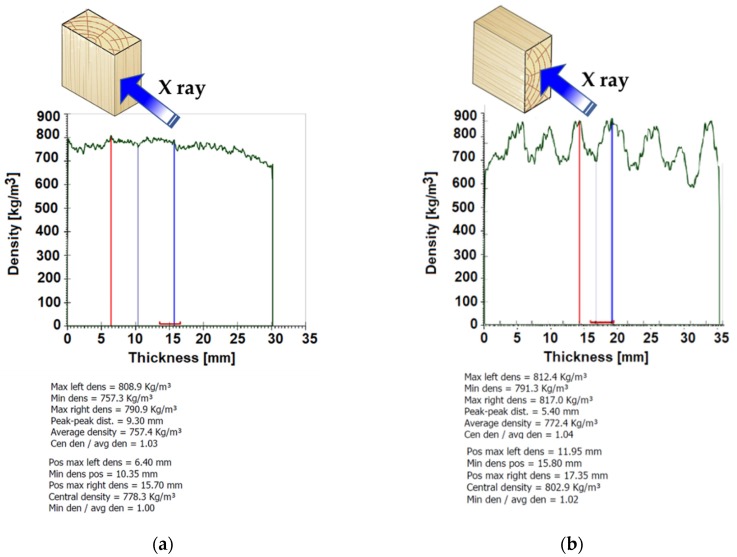
Variation of density profile of black locust wood samples: (**a**) longitudinal radial direction; (**b**) transversal direction of wood.

**Figure 7 polymers-12-00377-f007:**
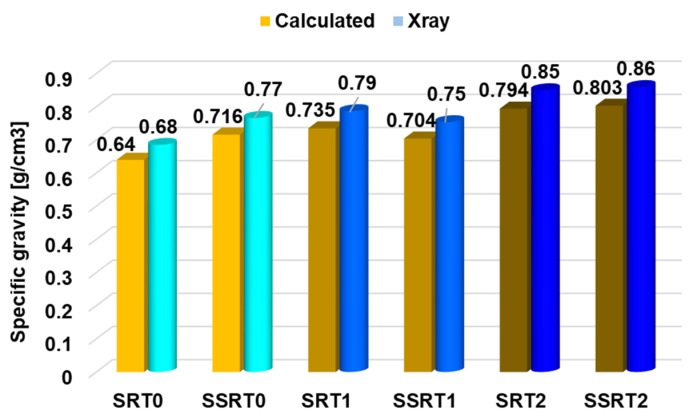
Comparison of conventional density (calculated) and density determined with X-ray equipment.

**Figure 8 polymers-12-00377-f008:**
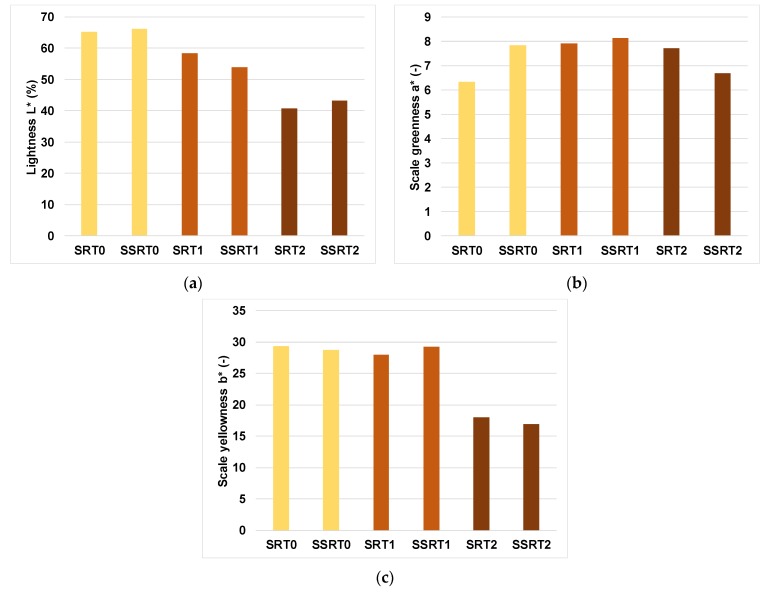
The colour measurement: (**a**)the lightness; (**b**) the scale greenness *a**; (**c**) the scale yellowness *b**.

**Figure 9 polymers-12-00377-f009:**
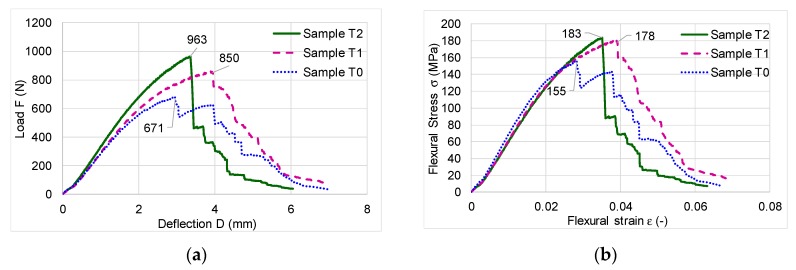
The characteristic curves to static three-point bending test (**a**) load versus deflection variation; (**b**) stress versus flexural strain variation.

**Figure 10 polymers-12-00377-f010:**
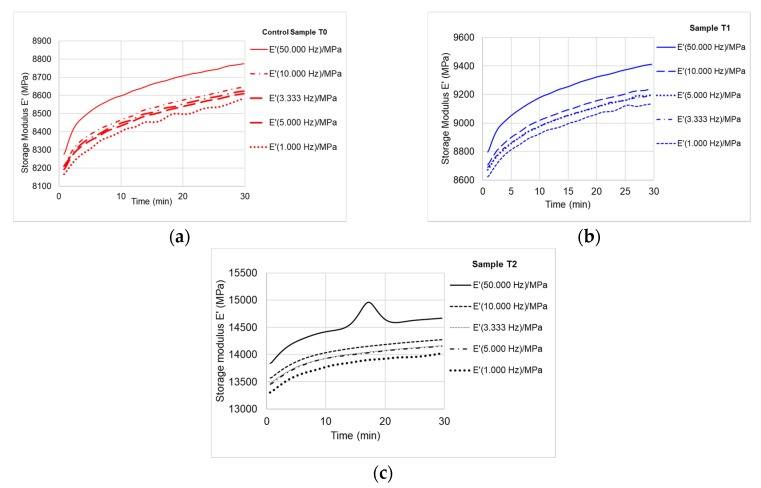
Variation of storage modulus *E*’ with loading frequencies: (**a**) control sample; (**b**) sample T1 (first treatment); (**c**) sample T2 (second treatment).

**Figure 11 polymers-12-00377-f011:**
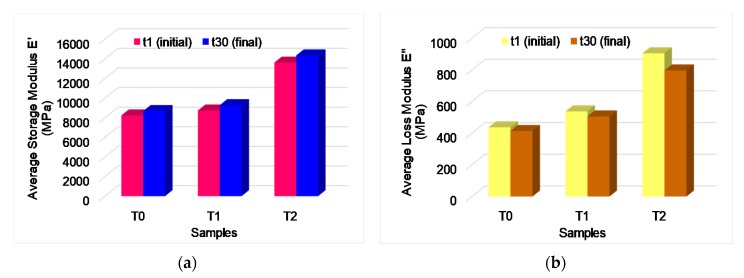
Comparison between dynamic properties of testes samples: (**a**) storage modulus *E*’; (**b**) loss modulus *E*”.

**Figure 12 polymers-12-00377-f012:**
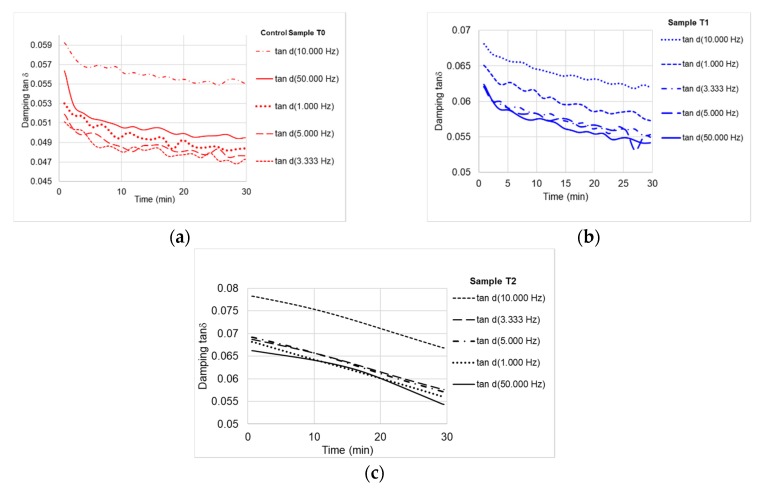
Variation of damping tanδ with loading frequencies: (**a**) control sample; (**b**) sample T1 (first treatment); (**c**) sample T2 (second treatment).

**Figure 13 polymers-12-00377-f013:**
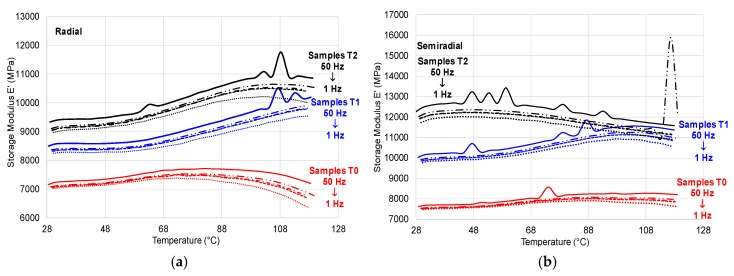
Variation of storage modulus *E*’ versus temperature: (**a**) samples cut in radial–longitudinal direction; (**b**) samples cut in semi-radial–longitudinal direction. Legend: red line—untreated samples; blue line—samples exposed to first ammonia treatment; black line—samples exposed to the second ammonia treatment; solid line (50 Hz); long dash dot dot line (10 Hz); dash dot line (5 Hz); square dot line (3.3 Hz); round dot line (1 Hz).

**Figure 14 polymers-12-00377-f014:**
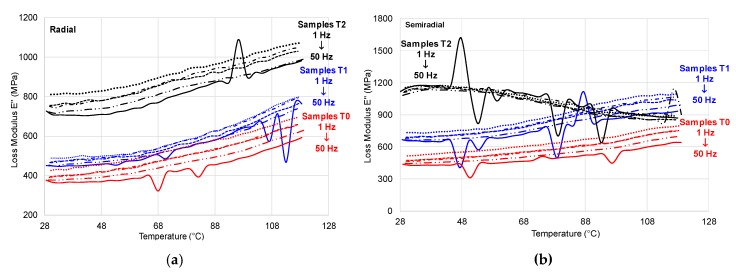
Variation of loss modulus *E*’’ with temperature: (**a**) samples cut in radial–longitudinal direction; (**b**)samples cut in semi-radial–longitudinal direction. Legend: red line—untreated samples; blue line—samples exposed to first ammonia treatment; black line - samples exposed to the second ammonia treatment; solid line (50 Hz); long dash dot dot line (10 Hz); dash dot line (5 Hz); square dot line (3.3 Hz); round dot line (1 Hz).

**Figure 15 polymers-12-00377-f015:**
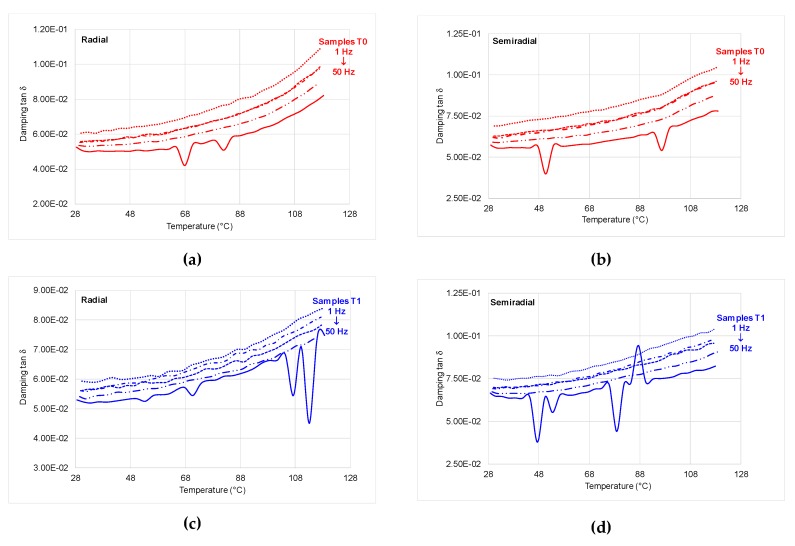
tanδ with temperature: (**a**) control sample—radial cross-section; (**b**) control sample—semi-radial cross-section; (**c**) sample T1 (first treatment)—radial cross-section; (**d**)—sample T1 (first treatment)—semi-radial cross; (**e**) sample T2 (second treatment)—radial cross-section; (**f**) sample T2 (second treatment)—semi-radial cross-section; solid line (50 Hz); long dash dot dot line (10 Hz); dash dot line (5 Hz); square dot line (3.3 Hz); round dot line (1 Hz).

**Figure 16 polymers-12-00377-f016:**
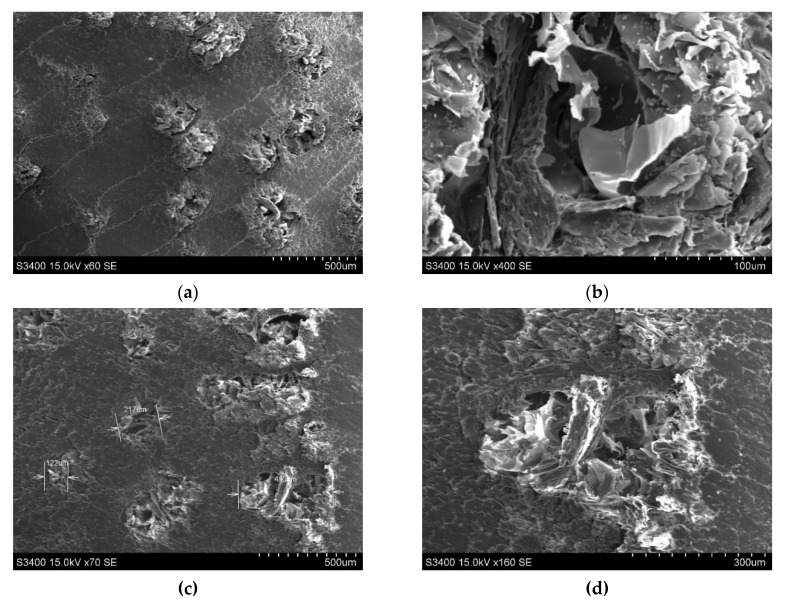
SEM view on cross-section of wood: (**a**) control sample magnification ×60 SE; (**b**) control sample magnification ×400 SE; (**c**) sample T1 (first treatment) magnification ×70 SE; (**d**) sample T1 (first treatment) magnification ×160 SE; (**e**) sample T2 (second treatment) magnification ×65 SE; (**f**) sample T2 (second treatment) magnification ×180 SE.

**Figure 17 polymers-12-00377-f017:**
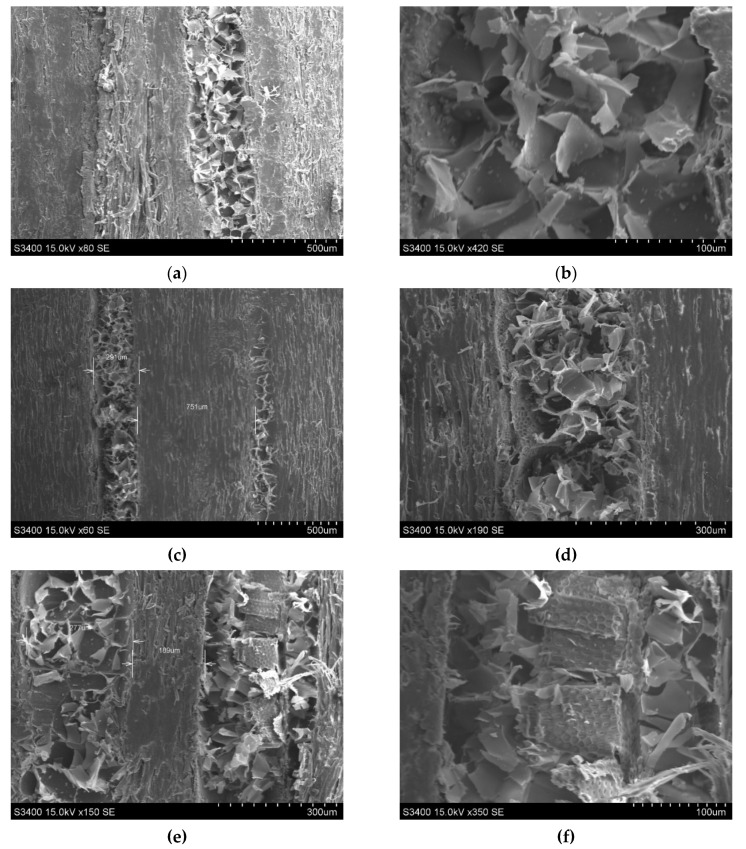
SEM view in longitudinal–radial section of wood: (**a**) control sample magnification ×80 SE; (**b**) control sample magnification ×420 SE; (**c**) sample T1 (first treatment) magnification ×60 SE; (**d**) sample T1 (first treatment) magnification ×190 SE; (**e**) sample T2 (second treatment) magnification ×150 SE; (**f**) sample T2 (second treatment) magnification ×350 SE.

**Table 1 polymers-12-00377-t001:** Physical characteristics of wood samples prepared for dynamic mechanical analysis (DMA) test.

Wood Sample Codes		Dimensions			Mass	Apparent Density	Average Density	Standard Deviation
		Width *b* (mm)	Length *L* (mm)	Thickness *h* (mm)	*m* (g)	*ρ* (g/cm^3^)	*ρ*_av_ (g/cm^3^)	STDV
S_R_T_0_ MC = 7%	S_R_T_0_1.	10.6	50	4.88	1.58	0.611	0.640	0.026
	S_R_T_0_2.	10.5	50	4.9	1.63	0.634		
	S_R_T_0_3.	10.56	49.95	4.92	1.73	0.667		
	S_R_T_0_4.	10.56	49.98	4.88	1.67	0.648		
S_SR_T_0_ MC = 7%	S_SR_T_0_1.	10.64	50.04	4.88	1.93	0.743	0.716	0.031
	S_SR_T_0_2.	10.64	50.05	4.86	1.92	0.742		
	S_SR_T_0_3.	10.6	50.03	4.86	1.79	0.695		
	S_SR_T_0_4.	10.6	50.01	4.88	1.77	0.684		
S_R_T_1_ MC = 7.5%	S_R_T_1_1.	10.35	49.9	4.97	1.94	0.756	0.735	0.021
	S_R_T_1_2.	10.59	50	5.02	1.88	0.707		
	S_R_T_1_3.	10.45	50.01	5.09	1.94	0.729		
	S_R_T_1_4.	10.45	49.98	4.94	1.93	0.748		
S_SR_T_1_ MC = 8%	S_SR_T_1_1.	10.49	49.96	4.92	1.63	0.632	0.704	0.085
	S_SR_T_1_2.	10.45	49.97	4.86	1.61	0.634		
	S_SR_T_1_3.	10.46	49.97	4.91	2.06	0.803		
	S_SR_T_1_4.	10.48	49.97	4.94	1.94	0.750		
S_R_T_2_ MC = 8%	S_R_T_2_1.	10.49	49.93	4.85	1.87	0.736	0.794	0.039
	S_R_T_2_2.	10.47	50.04	4.88	2.07	0.810		
	S_R_T_2_3.	10.58	50	4.85	2.10	0.819		
	S_R_T_2_4.	10.57	49.93	4.76	2.04	0.812		
S_SR_T_2_ MC = 8%	S_SR_T_2_1.	10.56	50	4.94	2.18	0.836	0.803	0.031
	S_SR_T_2_2.	10.55	50	4.96	2.08	0.795		
	S_SR_T_2_3.	10.52	50.01	4.92	2.11	0.815		
	S_SR_T_2_4.	10.52	49.99	4.98	2.00	0.764		

**Table 2 polymers-12-00377-t002:** Mechanical properties obtained from the static flexural test, due to the presence of ammonia in the black locust wood samples.

Average Values (10 Samples for Each Type)	Samples					
	T0	STDEV	T1	STDEV	T2	STDEV
MOE of Bending (MPa)	7190	236.44	6469	214.38	6338	268.53
Differences MOE (%)	0		−10.03		−11.85	
Break Force F (N)	679.19	84.64	860.93	103.80	963.50	110.74
Differences F (%)	0		+26.75		+41.86	
MOR of Bending (MPa)	156.51	24.63	180.51	25.14	183.41	28.53
Differences MOR (%)	0		+15.33		+17.18	
Maximum Deflection (mm)	7.01	0.512	7.00	0.653	6.05	0.721

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
