# Peer review of "Physical and Mechanical Properties of Ammonia-Treated Black Locust Wood"

_polymers, 2020, doi:10.3390/polym12020377_

Round 1

Reviewer 1 Report

The article studied the density, color change, and dynamic mechanical properties of ammonia treated acacia wood, trying to build up some connection between ammonia treatment and mechanical properties. DMA is one of the major techniques used in this article. However, it seems to have some concept error in the manuscript with regard to experimental design and result discussion. Also the presentation is confusing at certain part. DMA results were listed showing increased storage modulus with further ammonia treatment but no more insights was given. In addition to DMA test, some static test should serve better here as the author mentioned instrument application. For example, a destructive 3-point bending test to display the force, or tensile test showing young’s modulus and tensile strength.

It might be helpful to briefly introduce the applications of Acadia wood to the author of Polymers. Relative study targeting at those applications would be more appropriate. Page 4, 2.2.2. If an equation is present here, please explain the parameters like a0, a1, etc. Page 4, 2.2.3. The authors described here about their DMA design. From later Figures, it seems to be frequency scan at two cases: isothermal and temperature scan. But in Figure 9, the results seem to be obtained from an oscillation at fixed frequency rather than scan. My first question is, what is the purpose of doing frequency scan and temperature scan at the same time? Figure 10 gives Tg results, but as kinetic properties Tg is related with frequency, then what is the frequency here to obtain those TgDelta curve? Or it is a frequency scan at the same time? That will not be a good glass transition determination. Page 7, 3.3. The author explains the storage energy increasing with strain hardening. It does not look right to me.

Author Response

First we would like to thank the reviewers for carefully going through the manuscript and providing helpful suggestions for its improvement. Thanks to their constructive comments, we are able to present clearly and better version than the original manuscript. All the comments of the reviewers have been considered. In particular, the following changes have been made according to the reviewers' suggestions, highlighted by yellow colour in the manuscript.

Response to Referee 1

 Referee's comments:

The article studied the density, colour change, and dynamic mechanical properties of ammonia treated acacia wood, trying to build up some connection between ammonia treatment and mechanical properties. DMA is one of the major techniques used in this article. However, it seems to have some concept error in the manuscript with regard to experimental design and result discussion. Also the presentation is confusing at certain part. DMA results were listed showing increased storage modulus with further ammonia treatment but no more insights was given. In addition to DMA test, some static test should serve better here as the author mentioned instrument application. For example, a destructive 3-point bending test to display the force, or tensile test showing young’s modulus and tensile strength. It might be helpful to briefly introduce the applications of Acacia wood to the author of Polymers

We thank the reviewer for this relevant and thoughtful remark in the current context. The paper was rewritten taking into account the suggestions. The manuscript was extensively modified. The introduction was improved in accordance with suggestions and more references were aided. Also, in cap 2.3.3. we aided the destructive 3-points bending test on three types of samples which was performed using the universal machine LS100 Lloyd’s Instrument. We reanalysed the experimental results and we introduced them in new charts in accordance with your suggestions.

The acacia wood or black locust wood (Latin name Robinia pseudoacacia) has different applications as fence posts, boatbuilding, flooring, furniture, mine timbers, railroad ties, turned objects, fingerboards and back plate of guitars, some parts of musical instruments, veneer, wooden garden benches, pergolas, etc. For product diversification, harmonizing the colour of the wood with the architecture of the space and increasing the resistance to environmental factors, the colour of the acacia wood is modified by various treatments, but the most significant are chemical treatments.  

Page 4, 2.2.2. If an equation is present here, please explain the parameters like a0, a1, etc

We are thankful for this useful recommendation which makes the paper more understandable. We explain all parameters in revised manuscript:

Where  represents the overall colour change between samples treated with treatment T1 and untreated samples (T0);  - the overall colour change between samples treated with treatment T2 and samples treated with treatment T1;  - the overall colour change between samples treated with treatment T2 and untreated samples (T0);  - colour system in terms of lightness, greenness and yellowness of control samples (untreated);  - lightness, greenness and yellowness of treated samples with first treatment (5% aqueous solution of ammonium hydroxide);  - lightness, greenness and yellowness of treated samples with second treatment (10% aqueous solution of ammonium hydroxide);

 Page 4, 2.2.3. The authors described here about their DMA design. From later Figures, it seems to be frequency scan at two cases: isothermal and temperature scan. But in Figure 9, the results seem to be obtained from an oscillation at fixed frequency rather than scan. My first question is, what is the purpose of doing frequency scan and temperature scan at the same time? Figure 10 gives Tg results, but as kinetic properties Tg is related with frequency, then what is the frequency here to obtain those Tg Delta curve? Or it is a frequency scan at the same time? That will not be a good glass transition determination. Page 7, 3.3. The author explains the storage energy increasing with strain hardening. It does not look right to me.

We thank the reviewer for this excellent comment. We reorganized the presentation of the test steps at the DMA, as well as the way of the results representation, especially those where the temperature was scanned. We introduce the variation charts of the storage modulus, loss modulus and damping with temperature and for different loading frequencies. In new version of manuscript, we highlighted the major modifications and added parts in yellow colour.

Reviewer 2 Report

The work explores the effect of ammonia hydroxide treatment of acacia wood on its physical and mechanical properties – the issue having a certain impact on the wood technology. The strength of the work is that the authors made quite well justified and thorough study using a set of appropriate preparation and characterization techniques, enabling to get convincing novel data in this specific subject. The weakness is that the work lacks information on how the chemical composition of the wood changes upon treatment and how it correlates with the mechanical and physical properties. Nevertheless, the work presents enough data of appropriate level, is well arranged, accurately prepared, supported by relevant reference list, and, thus, can be published in Polymers with account of minor corrections:
1)    There is no indication of how charging of the insulating sample was overcome in SEM.
2)    Authors are to be consistent in spelling, i.e.:
lines 14, 145, etc. – “color”
lines 22, 236 – “colour”, etc

Author Response

Ref. Polymers - 697779

Physical and mechanical properties of ammonia treated black locust wood

Mariana D. Stanciu1,*, Daniela Șova1, Adriana Savin2, Nicolae Iliaș3 and Galina A.Gorbacheva4

Authors' Amendments

First we would like to thank the reviewers for carefully going through the manuscript and providing helpful suggestions for its improvement. Thanks to their constructive comments, we are able to present clearly and better version than the original manuscript. All the comments of the reviewers have been considered. In particular, the following changes have been made according to the reviewers' suggestions, highlighted by yellow colour in the manuscript.

Response to Referee 2

 Referee's comments:

The work explores the effect of ammonia hydroxide treatment of acacia wood on its physical and mechanical properties – the issue having a certain impact on the wood technology. The strength of the work is that the authors made quite well justified and thorough study using a set of appropriate preparation and characterization techniques, enabling to get convincing novel data in this specific subject. The weakness is that the work lacks information on how the chemical composition of the wood changes upon treatment and how it correlates with the mechanical and physical properties. Nevertheless, the work presents enough data of appropriate level, is well arranged, accurately prepared, supported by relevant reference list, and, thus, can be published in Polymers with account of minor corrections: 
1)    There is no indication of how charging of the insulating sample was overcome in SEM.

We thank the reviewer for this relevant and thoughtful remark in the current context. We mentioned in manuscript, in section 2.2.4. the procedure of preparation of wood samples for SEM and a figure with  coated samples.

Before SEM, the wood samples were coated with a thin layer of gold (Au) as a conducting material. The SEM was performed in semi-vacuum condition.

2)    Authors are to be consistent in spelling, i.e.: lines 14, 145, etc. – “color”; lines 22, 236 – “colour”, etc

We are thankful for this useful recommendation which makes the paper more understandable. We use only the term ‘’colour’’ in all manuscript. We also reorganized the presentation of the test steps at the DMA, as well as the way of the results representation, especially those where the temperature was scanned. In new version of manuscript, we highlighted the major modifications and added parts in yellow colour.

Round 2

Reviewer 1 Report

Thanks the authors for providing more info about bending test result. One question is, how many repeats for each group of sample? As those kind of mechanical properties is based on statistic not a single specimen. If multiple samples were tested for T0, T1 and T2, please show the error bar or standard deviation.

Author Response

Response to Reviewer 1 - minor revision (2nd response)

Thank you for useful recommendation. We specified in ch. 2.2.3. 1 the fact that 10 samples of each type were tested (we highlighted with green color). Then we have introduced in Table 2 the values of the standard deviation for each  mechanical property that we determined (highlighted in green color). 

Sincerely,

Mariana Stanciu
